



# Alkalinity and nitrate dynamics reveal dominance of anammox in a hyper-turbid estuary

Mona Norbisrath[1,2], Andreas Neumann[1], Kirstin Dähnke[1], Tina Sanders[1], Andreas Schöl[3], Justus E. E. van Beusekom[1], and Helmuth Thomas[1,2]

[1]Institute of Carbon Cycles, Helmholtz-Zentrum Hereon, Geesthacht, 21502, Germany
[2]Institute for Chemistry and Biology of the Marine Environment, Carl von Ossietzky University Oldenburg, Oldenburg, 26129, Germany
[3]Department of Microbial Ecology, Federal Institute of Hydrology, Koblenz, 56068, Germany

*Correspondence to*: Mona Norbisrath (mona.norbisrath@gmail.com)

## Abstract

Total alkalinity (TA) regulates the oceanic storage capacity of atmospheric $CO_2$. TA is produced along two general pathways, weathering reactions and anaerobic respiration of organic matter, e.g., by denitrification, the anaerobic reduction of nitrate ($NO_3^-$) to elemental nitrogen ($N_2$). Anammox, is another anaerobic pathway, yields $N_2$ as its terminal product via comproportionation of ammonium ($NH_4^+$) and nitrite ($NO_2^-$); this is, however, without release of alkalinity as a byproduct. In order to investigate these two nitrate / nitrite respiration pathways and their resulting impact on TA generation, we sampled the highly turbid estuary of the Ems River, discharging into the North Sea in June 2020. We sampled a transect from the Wadden Sea to the upper tidal estuary, five vertical profiles during ebb tide, and fluid mud for incubation experiments in the hyper-turbid tidal river. The data reveal a strong increase of TA and DIC in the tidal river, where stable nitrate isotopes indicate water column denitrification as the dominant pathway. In the fluid mud of the tidal river, the TA data imply only low denitrification rates, with the majority of the $N_2$ being produced by anammox (> 90 %). The relative abundances of anammox and denitrification, respectively, thus exert a major control on the $CO_2$ storage capacity of adjacent coastal waters.

## 1 Introduction

Global and climate change leads to profound challenges in the coastal zone. Amongst these are respective impacts on the $CO_2$ storage capacity of the coastal seas, which has gained increasing scientific attention (Burt et al., 2016;Reithmaier et al., 2020;Thomas et al., 2009;2007;Van Dam et al., 2021;Voynova et al., 2019). In contrast to the open ocean, the carbon cycle in coastal oceans generally exhibit much higher dynamics (Borges et al., 2005;Frankignoulle et al., 1998;Thomas et al., 2009;2004). These dynamics are even more accentuated in estuaries that are subject to human interventions like dredging, damming, nutrient inputs, or transportation.

Continental and intertidal estuaries, such as the Ems Estuary (German-Dutch border) on which we focus in this study, are the interface between coast and land. The Ems is exposed to high nutrient inputs, hosts high suspended particulate matter (SPM)





concentrations (Van Beusekom and De Jonge, 1998), and has shown already active nitrogen alternation (Sanders and Laanbroek, 2018;Schulz et al., 2022) which makes it a perfect natural laboratory to study the linkage of TA and nitrate respiration.

One of the largest impacts on the ocean is the increasing anthropogenic $CO_2$ content in the atmosphere, part of which is
absorbed by the ocean. The ocean's total alkalinity (TA), buffers the ocean's acidification by the absorbed weak acid $CO_2$. TA in turn is either generated by chemical rock weathering (Berner et al., 1983;Meybeck, 1987;Suchet and Probst, 1993), or metabolically by various anaerobic or inorganic pathways releasing alkalinity in various stoichiometries (Brewer and Goldman, 1976;Chen and Wang, 1999;Hu and Cai, 2011;Wolf-Gladrow et al., 2007).

Next to its role as a $CO_2$ buffer, the coastal zone is also exposed to high agricultural nutrient loads, which can lead to coastal
eutrophication, foster primary production, and increase seasonal anoxia (Große et al., 2016;Howarth et al., 2011;Nixon, 1995). These high nutrient loads also trigger nitrate respiration pathways. Potential pathways to remove nitrate and nitrite are denitrification and anammox. During denitrification, nitrate ($NO_3^-$) is reduced to nitrous oxide ($N_2O$) and then finally to dinitrogen gas ($N_2$). During denitrification, anaerobic bacteria, i.e. denitrifiers, use $NO_3^-$ as terminal electron acceptors and generate alkalinity in a ratio of 0.9 (Chen and Wang, 1999).

Unlike denitrification, anammox, the anaerobic ammonium oxidation, has no effect on TA (Middelburg et al., 2020). Here, the electron donor ammonium ($NH_4^+$) is oxidized with the electron acceptor nitrite ($NO_2^-$) to release $N_2$ (Meyer et al., 2005;Mulder et al., 1995;Thamdrup and Dalsgaard, 2002).

In light of governing the $CO_2$ buffer capacity of coastal oceans, the proportions in which anammox and denitrification serve to reduce $NO_2^-$ or $NO_3^-$ play a crucial role. We here combine carbon and nitrogen cycle observations to shed light on the
interaction between TA and respective nitrate or nitrite respiration pathways. To the best of our knowledge, this is the first time that alkalinity generation is used as a tool to discriminate and underpin calculated potential $N_2$ production pathways, and that anammox is found to dominate in such a eutrophic environment.

## 2 Materials and methods

### 2.1 Study site and water sampling

The Ems Estuary is located at the border between Netherlands and Germany. Including the outer estuary, it is approximately 100 km long, starts at the weir in Herbrum and discharges in the Wadden Sea (Fig. 1), which connects to the North Sea (NE Atlantic shelf). The tidal range increased from 2 m to 3.6 m (1950-2010) in the upstream estuary (de Jonge et al., 2014;van Maren et al., 2015). The Ems Estuary permanently underlies natural and anthropogenic pressures. Due to estuarine and riverine sea ports and the big German shipyard in Papenburg, the accessibility for large ships is needed. Channel deepening and
dredging activities that already started in the 1950s in the outer estuary and in the 1980s in the tidal river lead to hyper-turbid conditions (de Jonge et al., 2014). This topographic change results in increasing loads of marine suspended particulate matter (SPM), which accumulate in the upstream estuary and cause high oxygen demand for bacterial degradation and





remineralization (Jonge, 1983;de Jonge et al., 2014). Another indirect sink for oxygen are the high loads of fertilizer, which also increase eutrophication and oxygen consumption (Howarth et al., 2011;Nixon, 1995).

A special characteristic of the present Ems Estuary is the section of fluid mud in the bottom layers of the tidal river that reaches on average the upper 40 km of the estuary (Becker et al., 2018;de Jonge et al., 2014;Talke et al., 2009). Fluid mud is highly concentrated in SPM and separated from the overlying pelagic water.

Transect water samples were collected on board the RV *Ludwig Prandtl* (LP20200602) along the salinity gradient, starting from the North Sea northwest of the barrier island Borkum, to the weir in Herbrum during ebb tide on June 11th and 12th 2020

(Fig. 1). We continuously collected surface water samples (1.2 m depth) using a bypass from the onboard flow-through FerryBox system (Petersen et al., 2011), which provided important physical parameters such as salinity, temperature, and oxygen. For initial river values, we took one sample upstream of the weir. A special focus is placed on the area of the tidal river that reaches from the weir Herbrum (at Ems stream km -14.1) downstream to Ems stream km 36.

For incubation experiments, we sampled fluid mud with the bottom water sampler at two stations (Ems stream km 7.8 and

17.3, named as station 714 & 715, respectively) at ebb tide on June 11th 2020.

Additionally, vertical profile samples during ebb tide were provided by the German Federal Institute of Hydrology (Bundesanstalt für Gewässerkunde - BfG). These samples were taken at Ems stream km 7.2 on June 17th, six days after the surface transect sampling. Samples were taken from high tide to low tide, i.e. during ebb tide, with vertical samplings started at 11:01 (named as VP1), 12:29 (VP2), 13:58 (VP3), 15:00 (VP4), and 18:22 (VP5). Vertical profiles were measured at various

depths, either directly with probes (salinity, temperature, oxygen, depth), or indirectly by discrete water samples (nutrients, isotopes). The deepest samples were always taken 0.5 m above the bottom. Suspended particulate matter (SPM) was determined gravimetrically (DIN 38 409-H2). Water samples of 0.2 to 1 L volume were filtered through pre-sealed Whatman filters, dried at 550 °C, and weighed. Oxygen concentrations, salinity, temperature and depth were measured in situ with an YSI6660 EXO2 multi parameter probe.



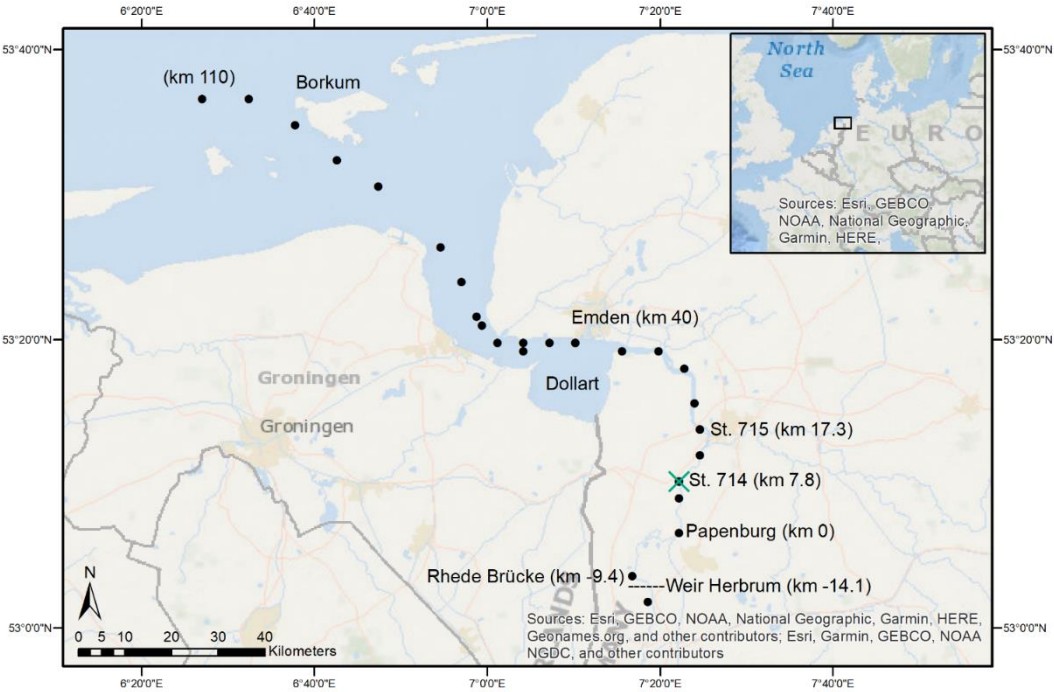

**Figure 1.** Sampling stations in the Ems Estuary in June 2020. The vessel based transect ended in Papenburg, corresponding to Ems stream km 0. We took samples at Rhede Brücke and Weir Herbrum from shore. The transect sampling stations are marked with the black dots, the vertical profile station at Ems stream km 7.2 with the green cross.

## 2.2 TA and DIC

For total alkalinity (TA) and dissolved inorganic carbon (DIC) measurements along the transect, we sampled 300 mL water with overflow into BOD (biological oxygen demand) bottles and preserved them with 300 µL saturated mercury chloride ($HgCl_2$) to stop biological activity. BOD bottles were closed air bubble free with ground-glass stoppers coated in Apiezon® type M grease, and secured with plastic caps. The samples were stored cool in the dark.

In the same way, we sampled two BODs out of the $N_2$ incubation experiment (see below) of both incubation stations for estimating TA generation rates in the fluid mud. We preserved the first BOD directly after sampling (0h) to get start values, i.e., initial concentrations of TA and DIC. We preserved the second BOD after 43 hours (43h) at the end of the incubation, to ensure a comparison between the start and end point of TA and DIC along the incubation (BOD incubation).

The parallel analyses of TA and DIC were carried out by using the VINDTA 3C (Versatile INstrument for the Determination of Total dissolved inorganic carbon and Alkalinity, MARIANDA - marine analytics and data), which measures TA by potentiometric and DIC by coulometric titration, respectively (Shadwick et al., 2011). To ensure a consistent calibration at both measurements, we used Certified Reference Material (CRM batch # 187) provided by Andrew G. Dickson (Scripps Institution of Oceanography).





### 2.3 Nutrients

Along the transect, water for nutrients was sampled with the FerryBox bypass, filtered over pre-combusted (4h, 450 °C) GF/F

filter, and stored frozen in 3 x 15 mL Falcon tubes.

The nitrate concentrations of the transect samples were analyzed with a continuous flow automated nutrient analyzer (AA3, SEAL Analytical) and a standard colorimetric technique (Hansen and Koroleff, 2007). Samples were analyzed in triplicate.

The water for nutrient samples of the vertical profiles was pumped onboard, centrifuged and the supernatant was measured back ashore. Samples of the vertical profiles were analyzed for nitrate, nitrite and ammonium with a continuous flow analyzer

(Skalar SC 13900) at the German Federal Institute of Hydrology (BfG). The automated procedure for determining ammonia is based on the modified Berthelot reaction (Krom, 1980). The determination of nitrate and nitrite is based on the Griess reaction and measured after Kroon (1993).

For calculating horizontal nutrient consumption or production rates, we used the following equation:

$$rate = \left(\frac{\Delta_{conc.}}{\Delta_{dist.}}\right) * \Delta_{time} \tag{1}$$

where the differences $\Delta$ of nutrient concentrations ($\Delta_{conc}$), time ($\Delta_{time}$), and bottom distance ($\Delta_{dist}$) between the two deepest depths of the vertical profiles VP4 and VP5 were used to detect a nutrient consumption or production during the end of ebb tide.

### 2.4 $\delta^{15}$N and $\delta^{18}$O stable isotopes

Along the transect, water for stable nitrate isotopes was sampled with the FerryBox bypass, filtered over pre-combusted (4h,

450 °C) GF/F filter, stored frozen in 100 mL PE bottles (acid-washed with 10 % HCl overnight), and analyzed by Schulz et al. (2022). Vertical profile water samples provided by BfG were also analyzed for stable nitrate isotopes by using the denitrifier method (Casciotti et al., 2002;Sigman et al., 2001). Briefly, the denitrifier method is based on a bacterial strain of *Pseudomonas aureofaciens* (ATCC#13985) that lacks nitrous oxide ($N_2O$) reductase activity and reduces nitrate and nitrite only to nitrous oxide and not further to $N_2$. The product is then measured using an isotope mass spectrometer (Delta Plus XP, Fisher Scientific)

coupled to a GasBench II.

In order to support the identification of pathways (e.g., Kendall et al., 2007), we calculated isotope effects ($\varepsilon$) for the deepest depth of the two last vertical profiles (VP4 & VP5), i.e. at the end of ebb tide, which showed the strongest nitrate decrease. We used an open-system approach assuming a steady state, and continuously supplied and partially consumed substrate, resulting in a linear relationship between the isotope values and the remaining fraction (*f*). The slope of the regression line

corresponds to the isotope effect ($\varepsilon$), and (*f*) to the remaining fraction of substrate at time of sampling (Sigman et al., 2009).

$$\varepsilon_{substrate} = \frac{\delta^{15}N \ (or \ \delta^{18}O) \ _{substrate} - \delta^{15}N \ (or \ \delta^{18}O) \ _{initial}}{(1-f)} \tag{2}$$

where the $\delta^{15}N \ (or \ \delta^{18}O)_{substrate}$ is the delta value at the time of sampling, and $\delta^{15}N \ (or \ \delta^{18}O)_{initial}$ is the initial delta value at the surface in the beginning of each vertical sampling.





$$f = \frac{[1 - C]}{[C_{initial}]} \qquad\qquad (3)$$

where $C$ is the concentration of nitrate at time of sampling and $C_{initial}$ is the initial nitrate concentration at surface, in the
beginning of each vertical sampling.

In addition, we measured stable isotopes of nitrite after the method by Böhlke et al. (2007), by using a bacterial strain of
*Stenotrophomonas nitritireducens* (ATCC#BAA-12) that reduces nitrite to nitrous oxide. We used these data in order to
quantitatively identify whether nitrite stems from organic matter (OM) and ammonium by nitrification, or from nitrate by

denitrification.

## 2.5 $N_2$ determination

### 2.5.1 Incubation set-up

For the incubation experiment, we took fluid mud samples with a bottom water sampler at ebb tide at two stations low in
oxygen, and filled them in one-liter Schott bottles without headspace. To get initial concentrations, we took samples before

adding the $^{15}N\text{-}NO_3^-$ tracer, with 137 µmol $NO_3^-$ $L^{-1}$ at station 714, and 144 µmol $NO_3^-$ $L^{-1}$ at station 715. After adding 30 mL
10 mM $^{15}N\text{-}NO_3^-$ tracer ($\geq$ 98 atom % $^{15}N$) stock solution, we closed the bottle bubble free with a rubber stopper for gentle
mixing. For sampling, the rubber stopper was prepared with two cut pipettes; one to fill the exetainer (12 mL Labco
Exetainer®), the other one to replace the outflowing water by air. The incubation ran in parallel replicates in exetainers for 1,
4, 18, 28 and 43 hours in total. The incubation time was adjusted to a parallel oxygen measurement by using a fiber-optic

sensor and FireSting set up (PyroScience GmbH) that started an hour after labelling (at t1). To inhibit the incubation and
preserve the samples, we injected approximately 100 µL 50 % zinc chloride ($ZnCl_2$) through the septum into each exetainer.

### 2.5.2 $N_2$:Ar analysis with the MIMS

In order to determine the $N_2$ production and to detect the different isotopic $N_2$ species $^{28}N_2$, $^{29}N_2$ and $^{30}N_2$ in the bottom water
of the Ems Estuary, we combined the $N_2$:Ar ratio measurements using the MIMS (Membrane Inlet Mass Spectrometer) (Kana

et al., 1994) with the expanded (Risgaard-Petersen et al., 2003) isotope pairing technique (IPT) (Nielsen, 1992). The addition
of the IPT allowed us to discriminate between denitrification and anammox as both pathways producing $N_2$ (Risgaard-Petersen
et al., 2003) and calculate potential $N_2$ production rates.

Samples for $N_2$ measurements were analyzed with a GAM-200 Quadrupol Mass Spectrometer (InProcess Instruments), which
was connected to an in-house-built flow-through membrane inlet and a cryo-trap (liquid $N_2$) to remove water vapor from the

sample stream. The stable nitrogen isotopes were measured at mass to charge ratio (m/z) 28 for $^{28}N_2$, at m/z 29 for $^{29}N_2$, and
m/z 30 for $^{30}N_2$, respectively. Argon was measured as the inert reference at m/z 40 (Kana et al., 1994). The integration time
was 0.5 s. For calculating the $N_2$ concentrations, we used equilibrium values calculated after Hamme & Emerson (2004). We
calibrated the sample measurements by using 3 different seawater standards (salinity of 20, 30, and 40) and MilliQ (0 salinity)
equilibrated at 5 °C.





We calculated the various produced parameters based on the equations by Risgaard-Petersen et al. (2003) in which D28 (denitrification of $^{28}N_2$) was calculated with their Eq. 17, D29 with Eq. 16, D30 equals $p^{30}N_2$ (the produced $^{30}N_2$ amount), A28 with Eq. 18, A29 with Eq. 19, P14 with Eq. 4, and r14 equals the ratio of measured initial $^{14}N\text{-}NO_3^-$ : added $^{15}N\text{-}NO_3^-$.

### 2.6 TA and N₂ production coupling

    In order to shed light on the interaction between TA and denitrification, we combined the TA generation of the BOD incubation

of each station with the average $N_2$ production per station. We used the average TA increase of the BOD incubations, equalized it with the maximum possible TA generation due to denitrification, and subtracted it from the total $N_2$ production. For equalize TA generated by denitrification, we calculated the TA gain as the corresponding amount of nitrogen. For this, we used the ratio of 0.9 for TA generation by denitrification (Chen and Wang, 1999) and subtracted it from the total $N_2$ production (Table 2). We assumed that the remaining produced $N_2$ would be produced by anammox. By using this approach, we can estimate

and prove the results of the IPT calculation, and discriminate potential $N_2$ production pathways by denitrification and anammox.

### 3 Results

### 3.1 Carbon status in the estuary

    The distribution of total alkalinity (TA) and dissolved inorganic carbon (DIC) ranged from 2357 µmol TA kg$^{-1}$ and 2541 µmol

DIC kg$^{-1}$ upstream of the weir in Herbrum, over 2487 µmol TA kg$^{-1}$ and 2671 µmol DIC kg$^{-1}$ directly after the weir, to a maximum of 2662 µmol TA kg$^{-1}$ and 2858 µmol DIC kg$^{-1}$ at Ems stream km 7.8. After the maximum, TA and DIC decreased to values around 2394 µmol TA kg$^{-1}$ and 2176 µmol DIC kg$^{-1}$ in the North Sea (Fig. 2a). We observed maximum TA and DIC concentrations in the tidal river, and in a range between salinity 0 and 5. From the weir to the TA maximum at Ems km 7.8 (five sampling points), we observed a TA gain of 305 µmol kg$^{-1}$.

The concurrent range of salinity varied from freshwater (0.5 salinity) to a salinity of 32.31 in the North Sea. The temperature varied between 18 °C and 15 °C along the entire transect from the weir in Herbrum (freshwater) downstream to the North Sea. In the outer estuary between Ems stream km 37 and 110 (salinity > 15), TA and DIC slightly decreased with increasing salinity, indicating the stronger impact from the Ems also on downstream regions (Fig. 2a,b). The almost linear behavior of DIC and TA indicated that ongoing processes act proportionally on both species (Fig. 2c). We observed a clear deviation from linearity

upstream of Ems stream km 36, i.e., in the tidal river (Fig. 2d). The $R^2 = 0.49$ indicated that around 50 % of the DIC can be explained with TA, the remaining indicating other, likely oxic processes being responsible for the DIC increase. The DIC excess in the surface water of the tidal river indicated to OM respiration (Wang et al., 2016) with high $CO_2$ generation (Fig. 2d).

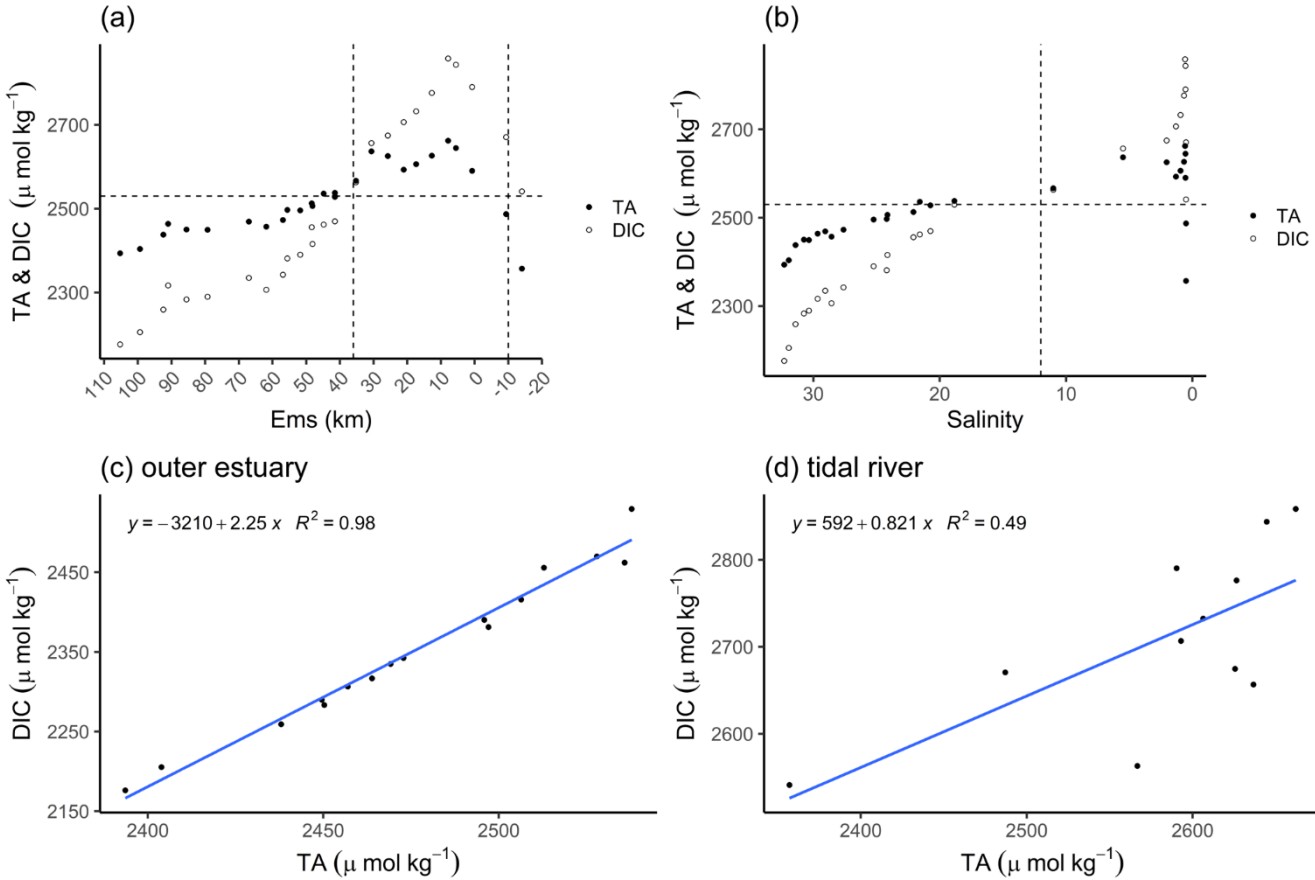


**Figure 2.** Total alkalinity (TA) and dissolved inorganic carbon (DIC) concentrations. The TA (black dots) and DIC (white dots) distribution is shown along the Ems Estuary in (a) km and (b) salinity. a) The Ems stream kilometer 110 is located around the barrier island of Borkum (North Sea), and the km -20 is upstream of the weir in Herbrum. The area between the weir and Ems stream km 36 is named as tidal river and visualized within the two vertical dashed lines. b) In the salinity plot, the tidal river is on the right side of the vertical dashed line between

salinity 0 and 12. The plots in the second row show the relation between TA and DIC of the transect samples in c) the outer estuary from Borkum to Ems stream km 36, and in d) the tidal river.

## 3.2 Nitrate assessment

### 3.2.1 Along the estuary

In a recent study, Schulz et al. (2022) identified a dominance of denitrification with nitrification as an additional contributor

in our focus area, the tidal river. They found $\delta^{15}$N values between 15 and 30 ‰ and suggested that these were caused by denitrification as the dominant process according to Kendall et al. (2007), visualized in Fig. 3a. Here, we can further distinguish the tidal river in two zones, in the upper tidal river (upstream the weir to Ems km 8) and in the lower tidal river (downstream Ems km 8 to Ems km 36). By using the data of Schulz et al. (2022), we observed a strong enrichment in the delta values in the

upper tidal river. Concurrently, we observed a maximum nitrate ($NO_3^-$) concentration upstream from the weir in the riverine

freshwater with 182 µmol $NO_3^-$ $L^{-1}$ decreasing downstream to 149 µmol $NO_3^-$ $L^{-1}$ at Ems km 7.8. This indicated a $NO_3^-$ loss of

32 µmol $L^{-1}$ in the upper tidal river where TA had the strongest increase to the maximum, and suggests denitrification as the

nitrate respiration pathway. In the lower tidal river, a mixed signal of denitrification and nitrification is visible. The lighter

delta values and the slightly increasing nitrate from Ems km 17 downstream, indicated the occurrence of nitrification. In the

salinity gradient, nitrate decreased to a common low marine concentration in the North Sea (Fig. 3a). Using the delta values,

we calculated the relationship between $\delta^{18}O$ and $\delta^{15}N$ ($\delta^{18}O$:$\delta^{15}N$) in the tidal river (Fig. 3b), which comes with 0.483 very

close to the slope of 0.5 (Kendall et al., 2007;Schulz et al., 2022) and supports the relevance of denitrification.

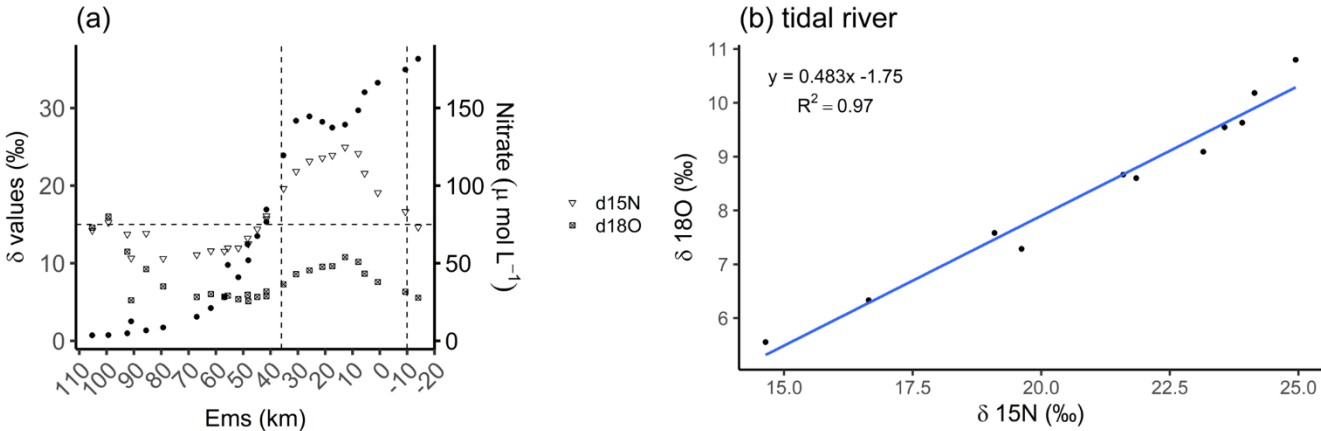

**Figure 3.** Nitrate based species distribution. a) The delta values of the nitrate stable isotopes $\delta^{15}N$-$NO_3^-$ (inverted triangle) and $\delta^{18}O$-$NO_3^-$

(crossed square) and the nitrate concentration (filled dots) along the Ems Estuary. b) The relationship between nitrate stable isotopes $\delta^{18}O$

and $\delta^{15}N$ in the tidal river (weir to Ems km 36). Data are from Schulz et al. (2022) who measured and analyzed the samples and data.

### 3.2.2 Vertical profiles

Six days after the cruise, samples of five vertical profiles (VP1 - VP5) were taken in the tidal river at Ems stream km 7.2 on

June 17[th] during ebb tide. The water temperature varied between 19.5 and 21.3 °C and decreased with increasing depths (Fig.

4h). In the freshwater, the salinity varied slightly between 0.41 and 0.45 (Fig. 4i).

The vertical differences in suspended particulate matter (SPM) leads to differences in density, which causes the occurrence of

vertical layering of the water masses. The SPM occurred in very high concentrations of above 40 g $L^{-1}$ near the bottom,

indicating that the fluid mud layer in the deepest 2 m is clearly separated from the overlying water (Fig. 4f).

Dual nitrate isotope values indicated strong fractionation in the deep layers, with $\delta^{15}N$-$NO_3^-$ values varying between 15 and 30

‰ and $\delta^{18}O$-$NO_3^-$ values between 5 and 15 ‰ (Fig. 4a,b). We observed highest $\delta^{15}N$-$NO_3^-$ values during ebb tide in the bottom

water. $\delta^{18}O$-$NO_3^-$ values increased strongly with depth in the first four samplings (VP1 - VP4), but we observed a lower level





of values towards the end of ebb tide. In particular, in the end of ebb tide (VP5), the isotope values of $\delta^{18}O$ were clearly lower than in the beginning, both from surface to bottom and over the tidal cycle towards the end of ebb tide.

Decreasing oxygen concentrations towards the bottom favor for denitrification with increasing depths (Fig. 4g). Anoxia in the

deepest layer facilitated anaerobic metabolic pathways such as denitrification and anammox, although there may be more pathways. Clearly decreasing nitrate concentrations indicate a loss of nitrate in the bottom layer during the entire ebb tide (Fig. 4c). The associated vertical profiles of nitrite and ammonium both showed an increase with depth during the ebb tide. Highest nitrite concentrations were found in bottom water (< 2 m) with values between 15 and 30 $\mu$mol $NO_2^-$ $L^{-1}$ (Fig. 4d). Ammonium concentrations were up to 5 $\mu$mol $NH_4^+$ $L^{-1}$ (Fig. 4e).

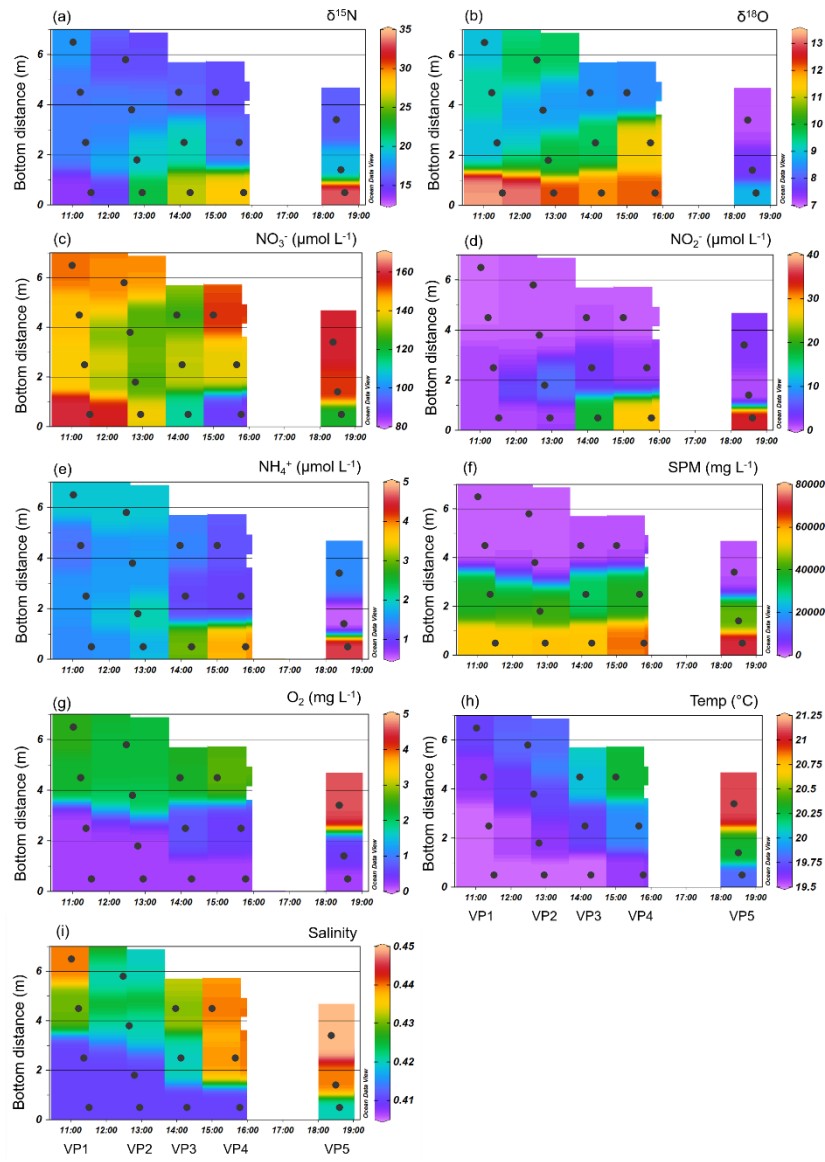






**Figure 4:** Vertical profiles of the various parameters during ebb tide at Ems stream km 7.2 on June 17[th] 2020. Delta values of the nitrate stable isotopes a) $\delta^{15}N$-$NO_3^-$ and b) $\delta^{18}O$-$NO_3^-$. Associated concentrations of c) nitrate, d) nitrite, e) ammonium, f) suspended particulate matter (SPM), g) oxygen, h) temperature, and i) salinity. Ebb tide ran from the left to the right side of each plot, indicated by decreasing water depths, which are visualized as bottom distance. The vertical samplings started at 11:01 (named as VP1), 12:29 (VP2), 13:58 (VP3), 15:00 (VP4), and 18:22 (VP5). The discrete samples are marked with dots, while the background values are gridded by using Ocean Data View.


Focusing on the two deepest depths with strongest nitrate loss of the vertical profiles VP4 and VP5, we were able to calculate average horizontal nutrient loss or production rates towards the end of ebb tide (Eq. 1). Considering the differences in nutrient concentration, distance, and time, we detected an average nitrate loss rate of -4 µmol $NO_3^-$ $L^{-1}$ $h^{-1}$ indicating an increasing nitrate consumption with ongoing ebb tide. At the same time, we determined average nitrite and ammonium production rates of 3.0 µmol $NO_2^-$ $L^{-1}$ $h^{-1}$ and 0.4 µmol $NH_4^+$ $L^{-1}$ $h^{-1}$, so that we observed higher nitrate loss than production of nitrite and ammonium.


In addition, we calculated the isotope effects ($\varepsilon$) of $\delta^{15}N$ and $\delta^{18}O$ (Eq. 2,3) as a linear regression for the two deepest depths of the vertical profiles VP4 and VP5 during ebb tide. The isotope effects ($\varepsilon$) of VP4 were 34.2 ‰ for $\delta^{15}N$ and 2.5 ‰ for $\delta^{18}O$, and of VP5 70.2 ‰ for $\delta^{15}N$ and 6.4 ‰ for $\delta^{18}O$.


In order to quantitatively narrow down potential sources of nitrite in the vertical profiles, we used the stable nitrite isotopes. These $\delta^{15}N$-$NO_2^-$ values revealed negative values in the first half of the ebb tide (VP1 - VP3) ($\delta$ values of -6, -7 ‰, and -0.2‰), but strongly positive $\delta$ values (between 10 and 22 ‰) in the second half of ebb tide (VP4 - VP5 ).


### 3.3 N₂ production via denitrification and anammox

The vertical profile data suggest a strong fractionation during nitrate respiration. To quantify nitrate respiration, we determined N₂ production in an incubation experiment of fluid mud from two stations in the tidal river.

Following Nielsen (1992), we identified average potential N₂ production rates of added $^{15}N$-$NO_3^-$ (D15) of 25.9 ± 6.1 µmol N $L^{-1}$ $h^{-1}$ and 19.3 ± 2.5 µmol N $L^{-1}$ $h^{-1}$ at stations 714 and 715, respectively (Table 1). At both stations, the N₂ production (D15) increased during the incubation, with a slightly higher production at the more upstream station 714 (Fig. 5a,b). For the N₂ production of in situ $^{14}N$-$NO_3^-$ (D14), we only detected an increasing N₂ production in the first 4 hours for station 714, and a relatively stable production over the incubation time for station 715. It appears that N₂ production from in situ $^{14}N$-$NO_3^-$ (D14) at station 714 increased first, and then decreased after the first hours due to depletion of the initial pool of the lighter isotope $^{14}N$ (Fig. 5b). Compared to station 715, we observed lower initial $NO_3^-$ concentrations at station 714.


In order to discriminate the relative contribution of denitrification vs. anammox in the incubations, we combined the N₂:Ar ratio measurements with the IPT after Risgaard-Petersen et al. (2003). Potential N₂ production rates were calculated with the linear regression of the respective concentration as a function of time during the incubation, as indicated by the slope (Fig. 5c-



h, Table 1). $N_2$ production increased with increasing incubation time via both anaerobic pathways. We calculated a potential
$N_2$ production by anammox of 10.16 to 22.95 µmol $L^{-1}$ $h^{-1}$, which is an order of magnitude higher than the production by
denitrification (0.10 to 1.01 µmol $L^{-1}$ $h^{-1}$, Table 2).

**Table 1.** $N_2$ production rates. $N_2$ production rates of added $^{15}N$-$NO_3^-$ (D15) and in situ $^{14}N$-$NO_3^-$ (D14) calculated after Nielsen (1992).
Denitrification rates of $^{28}N$ (D28), $^{29}N$ (D29) and $^{30}N$ (D30), anammox rates of $^{29}N$ (A29) and $^{28}N$ (A28), and genuine $^{14}N$-$N_2$ production
(P14) calculated after Risgaard-Petersen et al. (2003). The potential contribution of anammox to total $N_2$ production (ra) is presented as %.
Rates were calculated with the linear regression of the respective concentration as a function of the entire incubation time indicated by the
slope, which was used to calculate average values. The ra range shows the average minimum and maximum ra values of the incubation based
on the two replicates. Because we only had one initial $N_2$:Ar ratio for each station, we used this as a start value for the second replicate too.

| Species | Unit | Station 714 | Station 715 |
|---|---|---|---|
| **D15 N production (avg.) ± sd** | µmol $L^{-1}$ $h^{-1}$ | 25.9 ± 6.10 | 19.27 ± 2.5 |
| **D14 N production (avg.) ± sd** | µmol $L^{-1}$ $h^{-1}$ | -361.64 ± 147.5 | 430.78 ± 52.14 |
| **D29 N production (avg.) ± sd** | µmol $L^{-1}$ $h^{-1}$ | 1.01 ± 0.09 | 0.37 ± 0.12 |
| **D30 N production (avg.) ± sd** | µmol $L^{-1}$ $h^{-1}$ | 0.96 ± 0.09 | 0.33 ± 0.11 |
| **D28 N production (avg.) ± sd** | µmol $L^{-1}$ $h^{-1}$ | 0.27 ± 0.02 | 0.10 ± 0.03 |
| **A29 N production (avg.) ± sd** | µmol $L^{-1}$ $h^{-1}$ | 22.95 ± 5.83 | 18.28 ± 2.20 |
| **A28 N production (avg.) ± sd** | µmol $L^{-1}$ $h^{-1}$ | 12.12 ± 3.08 | 10.16 ± 1.21 |
| **P14 N production (avg.) ± sd** | µmol $L^{-1}$ $h^{-1}$ | 25.79 ± 6.30 | 20.90 ± 2.60 |
| **ra range (avg. min - max)** | % | 94.5 – 99.4 | 97.4 – 98.0 |







**Figure 5.** $N_2$ production during the incubation. $N_2$ production calculated after Nielsen (1992) of a) added $^{15}N$-$NO_3^-$ (D15) and b) in situ $^{14}N$-$NO_3^-$ (D14). $N_2$ production calculated after Risgaard-Petersen et al., (2003) via denitrification of c) $^{29}N_2$ (D29), d) $^{30}N_2$ (D30) e) $^{28}N_2$ (D28), and via anammox of e) $^{29}N_2$ (A29), and f) $^{28}N_2$ (A28). g) Estimation of the genuine $^{14}N$-$N_2$ production (P14) during the incubation after Risgaard-Petersen et al., (2003). Two replicates of each station are given, with station 714a (orange) and 714b (yellow), and station 715a (dark blue) and 715b (light blue). The slope of the linear regression indicates the production rate of the respective $N_2$ production as a function

of incubation time. Because we only had one initial $N_2$:Ar ratio for each station, we used this as a start value for the second replicate too.

### 3.4 TA incubation in BODs

In order to estimate the increase in TA in fluid mud and calculate potential TA generation rates, we used the BOD incubations. In these samples, TA was higher than in surface samples (Fig. 6), with initial concentrations for TA and DIC > 3000 µmol kg$^{-1}$. During the incubation, we measured increasing TA and DIC concentrations in both stations from the start (0h) to the end

(43 h). At station 714, TA increased from 3300 to 3378 µmol TA kg$^{-1}$ and DIC from 3337 to 3435 µmol DIC kg$^{-1}$. In contrast, station 715 had somewhat lower values with TA increasing from 3168 to 3196 µmol TA kg$^{-1}$ and DIC from 3303 to 3340 µmol DIC kg$^{-1}$. We used the increasing difference to calculate potential generation rates for station 714 of 1.8 µmol TA kg$^{-1}$ h$^{-1}$ and 2.3 µmol DIC kg$^{-1}$ h$^{-1}$, and for station 715 of 0.7 µmol TA kg$^{-1}$ h$^{-1}$ and 0.9 µmol DIC kg$^{-1}$ h$^{-1}$ (Table 2).

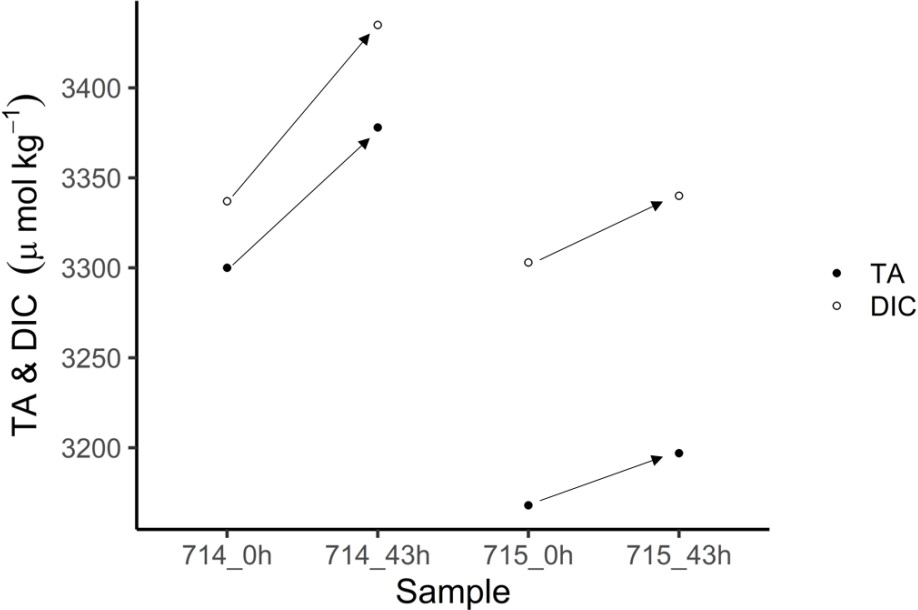

**Figure 6.** BOD incubations. The start (0h) and end (43h) concentrations of TA (black dots) and DIC (white dots) of the BOD incubation samples from station 714 and 715, respectively.





**Table 2.** C and N values of the incubation experiment. TA generation is the rate of TA generation in the BOD incubation. TA as $NO_3^-$ loss is the consumed $NO_3^-$ that correspond to the amount TA generated (equated with denitrification) recalculated with 0.9. D15 N production is the total produced amount of added $^{15}N$-$NO_3^-$ after Nielsen (1992) and calculated as the slope of a linear regression of the produced $^{15}N_2$ concentration as a function of the entire incubation time. Calc. denitrification N production and calc. anammox N production is the estimated amount of added $^{15}N$-$NO_3^-$ that was produced by denitrification and anammox, respectively. The estimation based on the approach of combining TA and the $N_2$ production calculation after Nielsen (1992). We assumed the $N_2$ produced by anammox as the remaining amount $N_2$ that was not produced by denitrification. The standard deviation as sample variability is given if possible as ± sd. Average values were calculated of two replicates.

| Species | Unit | Station 714 | Station 715 |
|---|---|---|---|
| **TA generation** | µmol kg$^{-1}$ h$^{-1}$ | 1.8 | 0.6 |
| **TA as $NO_3^-$ loss** | µmol L$^{-1}$ h$^{-1}$ | 2.0 | 0.7 |
| **Avg. D15 N production ± sd** | µmol L$^{-1}$ h$^{-1}$ | 25.9 ± 6.1 | 19.3 ± 2.5 |
| **Avg. Calc. denitrification N production (D15)** | % | 7.7 | 3.5 |
| **Avg. Calc. anammox N production (D15)** | % | 92.3 | 96.5 |
| **Avg. Calc. denitrification N production (D14)** | % | -0.6 | 0.2 |
| **Avg. Calc. anammox N production (D14)** | % | 100.6 | 99.8 |

## 4 Discussion

### 4.1 TA source in the estuary

In comparison to the North Sea with TA and DIC values ranging around 2400 µmol TA kg$^{-1}$ and 2100 µmol DIC kg$^{-1}$ (Thomas et al., 2004), and other German estuaries such as the Elbe Estuary with average values around 2000 µmol kg$^{-1}$ e.g., (Amann et al., 2015;Brasse et al., 2002;Norbisrath et al., 2022), the TA and DIC concentrations in the Ems Estuary were clearly higher with values between 2400 µmol kg$^{-1}$ and 2900 µmol kg$^{-1}$. In particular in the fluid mud, TA and DIC concentrations exceeded the water column concentrations of the Ems Estuary and adjacent zones with values > 3000 µmol kg$^{-1}$. Thus, the upper Ems Estuary and especially the fluid mud is a clear source of TA, probably driven by anaerobic pathways.

### 4.2 Nitrate respiration in the tidal river

Stable nitrate isotopes are used to identify nitrate sinks and sources (Kendall et al., 2007;Middelburg and Nieuwenhuize, 2001;Sigman et al., 2001). According to Kendall et al. (2007), $\delta^{15}N$ values between 15 and 30 ‰ suggest denitrification as the dominant pathway, which is an important sink for anthropogenic nitrate and the global nitrogen budget. In a recent study, Schulz et al. (2022) identified three distinct biogeochemical zones along the Ems Estuary with different predominant nitrogen cycling pathways based on water column properties and stable isotopes. In our focus area, the tidal river, Schulz et al. (2022) found a dominance of denitrification, with nitrification as an additional contributing pathway in the downstream part.





In addition to the assessment along the estuary, we investigated five vertical profiles at km 7.2 in the tidal river during ebb tide. They allowed us to scan and detect vertical differences in the water column. The hydrographic properties such as temperature, salinity, and SPM confirmed the clear stratification of the fluid mud in the Ems Estuary.

In accordance with the transect data, the observed vertical delta values of nitrate also indicate strong fractionation in all depths
supporting denitrification to occur in the water column. The occurrence of denitrification in the water column of the tidal river is also supported by the isotope effects ($\varepsilon$). The isotope effect of $\delta^{15}N$ (34.2 ‰) of VP4 came close to the range for water column denitrification that can vary between 20 and 30 ‰ (Sigman and Casciotti, 2001), the isotope effect of $\delta^{15}N$ (70.2 ‰) of VP5 was very high and exceeded this range by far. While we do not have an explanation for this unexpectedly high isotope effect, it is clear that strong fractionation acts on nitrate isotopes, and denitrification and anammox both are known to coincide
with significant effects (Brunner et al., 2013;Casciotti et al., 2003;Granger and Wankel, 2016;Mariotti et al., 1981;Sigman and Casciotti, 2001). Noticeable are the lower $\delta^{18}O$ values in both vertical and horizontal directions, with $\delta^{18}O$ isotope effects of 2.5 ‰ of VP4 and 6.4 ‰ of VP5, respectively. We observed a lower fractionation of $\delta^{18}O$ than $\delta^{15}N$. Interestingly, Dähnke and Thamdrup (2016) found that anammox activity can lead to a decoupling of N and O isotope effects, and that the oxygen isotope effect of nitrate consumption can even be reversed, which matches our findings. Thus, we speculate that the decrease
in $\delta^{18}O$ values might point towards anammox activity in the fluid mud.

The observed vertical loss of nitrate revealed a nitrate consumption due to water column denitrification and / or nitrate reduction to nitrite (Fig. 4c). High nitrate concentrations were present during the entire tide and provided sufficient substrate for $N_2$ producing pathways.

Nitrite concentrations increased with depth to maximum concentrations (> 30 µmol $L^{-1}$) in the deepest layer, which is the
intermediate product of either nitrification or denitrification. In the light of decreasing nitrate and oxygen concentrations in the water column, and anoxic conditions in the fluid mud, we rather expect nitrate respiration as the source of nitrite in the deeper layers. In addition, the vertical nitrate loss being higher than nitrite production suggests that nitrite production is fueled by denitrification of nitrate. In order to qualitatively investigate the nitrite sources and support our above results, we used the nitrite stable isotopes. The observed lighter, negative $\delta^{15}N$-$NO_2^-$ values in the first half during ebb tide point towards
nitrification (ammonia oxidation) as source for nitrite, which is an aerobic pathway occurring in the more oxygenated upper water layers during high tide. In contrast, the observed change from the earlier light $\delta^{15}N$-$NO_2^-$ in the first half during ebb tide to heavy, positive $\delta^{15}N$-$NO_2^-$ values in the anoxic deeper layers at nearly ebb tide indicates a change in reaction pathways. The heavy, positive $\delta^{15}N$-$NO_2^-$ values were similar to the observed δ values of nitrate, and are more indicative of nitrate and thus denitrification as the main source of nitrite. This newly produced nitrite is then accessible for either denitrification or anammox.
The combination of the high concentrations of nitrite and also ammonium in the deepest layer suggest OM degradation as a potential source of nitrite and ammonium (Blackburn and Henriksen, 1983). The occurrence of such high concentrations of nitrite and ammonium provide sufficient substrate for anammox. The strong stratification of the fluid mud with anoxic conditions facilitate the occurrence of anaerobic $N_2$ producing pathways in the bottom water of this highly turbid estuary.





We further performed an incubation experiment with fluid mud, and were able to discriminate $N_2$ production by denitrification
and anammox. When calculating $N_2$ production according to Risgaard-Petersen et al. (2003), we found that more than 90 %
of $N_2$ in the fluid mud at both stations and each time step (Table 2) must originate from anammox (ra).

**4.3 TA and $N_2$ production coupling**

The coupling of TA and $N_2$ production in the tidal river of the Ems Estuary can be used to locate the most active area for TA
generation and link part of the increase to water column denitrification. We were also able to identify nitrate / nitrite
consumption in the fluid mud and discriminate between denitrification and anammox. In the following, we combined TA
generation and nitrate respiration pathways first for the transect surface samples, second for the vertical profiles of the water
column, and third for the fluid mud.

Stable nitrate isotopes in surface samples of the present Ems Estuary transect revealed denitrification (Schulz et al., 2022) as
the dominant pathway in the inner estuary in the tidal river (weir to Ems stream km 36). The observed denitrification activity
matches with the observed increased TA values in this area. However, the general TA gain between upstream the weir and the
TA maximum at Ems stream km 7.8, the upper tidal river, was much higher than the concomitant nitrate loss. We assume that
parts of the TA gain were generated by denitrification, but also other metabolic anaerobic pathways or $CaCO_3$ dissolution
could generate TA in the water column of the tidal river of the Ems.

In the vertical water column profiles, we observed decreasing nitrate concentrations indicating nitrate consumption with depth.
In combination with the low oxygen concentrations that also decreased with depth, and the increasing delta values with depth
(Fig. 4), we suggest that denitrification appeared in the water column, in particular in the deeper layers, generating TA as a
byproduct. The decreasing nitrate and oxygen concentrations and the increasing delta values suggest that denitrification activity
highly and frequently occurred towards the bottom.

The fluid mud is indicated by strong vertical gradients of most observed parameters in the deepest 1 m and suggest a strong
separation from the upper water column. In particular in the last sampling (VP5), i.e., at strongest ebb tide, the substantial
nitrate consumption resulted in a significant increase of $\delta^{15}N$ values of the remaining nitrate pool. Ongoing denitrification
binds $H^+$ ions, thus increases TA values. We indeed observed higher TA concentrations in the fluid mud than in the water
column and suspect that at least some TA is generated by denitrification in the deepest layers.

In order to further shed light on the pathways in the fluid mud, we combined the BOD incubation with the $N_2$ production
experiment. Assuming that all TA generation in the BOD incubation containing fluid mud was due to denitrification, which
highly occurred in the deeper layers, we related the TA increase from the BOD incubation to $N_2$ generation according to
Nielsen (1992), and thus estimated $N_2$ generation by denitrification.

Given a TA generation by denitrification with a ratio of 0.9 (Chen and Wang, 1999), we calculated theoretical denitrification
rates based on TA generation and compared the results with total $N_2$ production of D15 based on Nielsen (1992) (Table 2).
We found that a large fraction of $N_2$ production could not be accounted to denitrification. Based on this mismatch, it appears
that 90 % of the $N_2$ production stems from anammox, which is neutral with regards to TA generation (Middelburg et al., 2020).





The dominance of $N_2$ production by anammox in the fluid mud (90 % $N_2$ production) is supported by the IPT calculation after Risgaard-Petersen et al. (2003), in $^{15}N$-$NO_3^-$ and $^{14}N$-$NO_3^-$.

It is intriguing that we find such a high contribution of anammox in a heterotrophic estuary. Traditionally, anammox has been
associated with deep sediments that are poor in organic matter and have low oxygen consumption rates (Dalsgaard et al., 2003;Engström et al., 2005;Kuypers et al., 2003;Thamdrup and Dalsgaard, 2002).

Other studies that focused on estuarine and coastal regions also found a clear contribution of anammox, albeit in a much lower range. For instance, in the Thamse Estuary (UK), the occurrence of anammox was observed to be much lower with a contribution to $N_2$ production of < 10 % (Trimmer et al., 2003). Similar low values were observed during summer in the
Randers Fjord in Denmark (Risgaard-Petersen et al., 2004), in estuarine and coastal sites in Rhode Island in the USA (Brin et al., 2014), and in the subtropical Logan river in Australia (Meyer et al., 2005). A two times higher maximum contribution of anammox to $N_2$ production (22 %) was observed by Rich et al. (2008) in Chesapeake Bay, USA.

But what can be the cause for the dominance of anammox that we find at our study site?

Anammox correlates with high concentrations of $NO_3^-$ in the overlying water and high organic carbon content (Nicholls and
Trimmer, 2009;Trimmer et al., 2003), which were both present in the tidal river of the Ems Estuary. Dalsgaard et al. (2012) also pointed out the correlation of higher anammox rates at stations with higher ammonium concentrations (close to 5 $\mu$mol $L^{-1}$) and relatively high turbidity in the work of Hamersley et al. (2007) and Lam et al. (2009). In the tidal river of the Ems, and in particular in the fluid mud (Fig. 4), we also observed high concentrations of ammonium, providing sufficient characteristics for anammox.

In addition, the reactivity of the OM can affect anammox (Engström et al., 2005;Trimmer et al., 2003). Engström et al. (2005) identified that in particular fresh and reactive OM favors denitrification, whereas old unreactive OM correlates with higher anammox rates. The reason is probably the difference in maximum growth rates, which favors fast-growing denitrifying bacteria over slow-growing anammox bacteria when available substrate is abundant (You et al., 2020 and references therein). Interestingly, Schulz et al. (2022) found high C:N ratios in this river section, and postulated a low organic matter reactivity
there. Thus, it is plausible that OM in the fluid mud is mostly refractory, because fresh imported OM is already degraded in the water column above the fluid mud, or in the middle reaches of the estuary before reaching the riverine part of the estuary. The DIC excess in relation to TA supports OM degradation (Wang et al., 2016) in the overlying water. This would be in line with the stratification that we find, which separates the fluid mud from the overlying water (Fig. 4). In this case, denitrifier growth and thereby denitrification may be limited by the low reactivity of the OM. This unique niche characterized by high
availability of nitrate, nitrite and ammonium, recalcitrant OM, and low oxygen enables anammox to gain importance in the Ems Estuary.

We showed that both denitrification and anammox occurred in the deeper layer of the tidal river, but suggest that anammox is the major $N_2$ producing pathway in the fluid mud.

## 5 Conclusion

In the highly turbid inner Ems Estuary, we observed TA and DIC concentrations being higher than in the North Sea with both values between 2400 and 2900 µmol kg$^{-1}$, indicating the estuary as a source for TA. Denitrification occurred in the water column of the tidal river, where we also detected the highest TA concentrations. In this area in the water column, the TA gain is much higher than the nitrate loss, suggesting that anaerobic pathways other than denitrification, or $CaCO_3$ dissolution could generate TA.

In the incubation experiment, denitrification rates as identified according to Risgaard-Petersen et al. (2003) were very low and anammox contributed > 90 % to potential $N_2$ production. Assuming all TA being generated by denitrification in the fluid mud, we were able to discriminate the $N_2$ production pathways in the high turbidity part of the estuary and underpin the $N_2$ production calculations according to Risgaard-Petersen et al. (2003). We found denitrification to occur in the deeper layers of the water column, but identified anammox as the major $N_2$ production pathway at least in the fluid mud of the tidal river in the Ems

Estuary.

*Data availability*

The data set of the study by Schulz et al. (2022) is available at https://doi.pangaea.de/10.1594/PANGAEA.942222. The carbon data are in preparation to be released in coastMAP Geoportal (https://www.coastmap.org) connecting to PANGAEA with doi, and are available upon request from the corresponding author.

*Acknowledgement*

We thank the crew from RV *Ludwig Prandtl* to enable the cruise, Leon Schmidt for the support during the cruise and the nutrient measurements, and Markus Ankele for the isotope measurements. Thanks to Wasserstraßen- und Schifffahrtsamt Ems-Nordsee for providing the Ems River kilometer data, and Linda Baldewein for helping with river kilometer calculations. This research has been funded by the German Academic Exchange Service (DAAD, project: MOPGA-GRI, grant no. 57429828),

which received funds from the German Federal Ministry of Education and Research (BMBF).

*Author contribution*

MN did the sampling, sample measurement and analyses, the data interpretation and evaluation. AN supported with the MIMS set up, the MIMS data preparation and interpretation. TS & JVB organized the cruise, supported the sampling and data interpretation. JVB did the river sampling at Rhede Brücke and Weir Herbrum. AS did the vertical profile sample collection,

measurements, data preparation, and method description. MN, KD, & HT designed this study. MN prepared the manuscript with contribution from all co-authors.



*Competing interests*

The authors declare that they have no conflict of interest.

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
