# Peer review of "Alkalinity and nitrate dynamics reveal dominance of anammox in a hyper-turbid estuary"

_Biogeosciences, 2022_

## Author Comment (AC1)

**RC1**: 'Comment on bg-2022-226', Anonymous Referee #1, 01 Jan 2023 reply

This paper focuses on nitrogen cycling, specifically nitrogen loss mechanisms, within an estuary. By supplementing nitrogen measurements with carbon system measurements, the researchers estimated the amount of fixed nitrogen loss that occurred due to canonical denitrification versus anammox. High resolution sampling from a river endmember to the North Sea, incubations, and a process station provide detailed characterization of the study site. Unfortunately, I struggled with following several analyses in this paper, and I would appreciate the authors clarifying these sections, which in many cases involves presenting the data in a different fashion. I suggest major revisions, as specified below.

*AC: Dear referee #1, we thank you for your feedback and the helpful comments and suggestions. We will address your suggestions in order to improve the manuscript.*

1. The primary conclusions of this paper hinge on comparing total alkalinity measurements and $N_2$ production rates in fluid mud samples. I did not entirely understand the conditions for these incubations as well as the specific mathematics for concluding that anammox is the dominant metabolism. Here are some specific challenges
   1. Line ~145: In the incubation set-up, the authors specify that fluid mud samples were taken from stations that were low in oxygen. Can you quantify its concentration? If the concentration is below the sensor detection limit, can you specify that and the limit you are using?

      *AC: Yes, we can specify the surface oxygen concentrations at this station, as well as the oxygen concentrations at the start of the incubations. We will add the data to the revised version.*

   2. On this same idea, anaerobic respiration also increases total alkalinity. It is not clear to me how this paper differentiates the amount of alkalinity produced by aerobic respiration versus canonical denitrification. Perhaps the oxygen concentration is so low that the small amount of alkalinity produced it produces is neglible? Aerobic respiration does produce a small amount of alkalinity (see "An assessment of ocean margin anaerobic processes on oceanic alkalinity budget" by Hu and Cai (2011) for the precise mathematics).

      *AC: Because of the very low oxygen concentrations ($<60$ $\mu M$ ($<2$ mg $L^{-1}$) at the start of the incubation and the decreasing oxygen concentrations during the incubation, we rather exclude aerobic respiration as source of alkalinity in the incubation experiment and the fluid mud.*

   3. The structure of Fig. 6 took me a long time to understand the information it contained. Perhaps a plot of DIC vs TA with two lines on it for each sample, and the caption saying that 43 hours elapsed? Another option could be two subplots of TA and DIC with time on the x-axis, as this could also be easier to parse. In addition, I think that it would be helpful to see the $N_2$ data over time for these incubations. Since these values are also averaged, can you add error bars? To visually represent the calculations performed and the link between alkalinity and $N_2$ production, perhaps another figure plotting TA vs $N_2$ and then adding in vectors for the denitrification-derived $N_2$ and annamox-derived $N_2$ could improve clarity?

*AC: Thank you for these suggestions. We can think about another style of plotting with time on the x-axis to better visualize the time elapse. We will also add a more precise description. A plot of DIC vs. TA is not useful since it would show another result. Because the TA/DIC and $N_2$ incubation were not in the same bottle, and we only have start and end values for the TA/DIC incubation, we think a combination of both as not useful, in particular, because the $N_2$ production of all $N_2$ species of these incubations is given in Fig. 5. Because we plotted each incubation $N_2$ pattern as a single line (Fig. 5), error bars are not necessary. However, the standard deviation of the averaged $N_2$ value that we used for this estimate is given in table 2.*

4. From the description of your methods (line 170), I anticipate that you calculated the amount of $N_2$ that would be produced based on the measured TA, then compared that TA-calculated $N_2$ against the measured $N_2$ to determine the annamox $N_2$. I would like some clarity that this is the method that was used, because some of the verb choice confused me.
    1. Saying "…we combined the TA generation… with the average $N_2$ production per station" suggests to me that these values were added. Would "compared" be a better verb than "combined"?
    2. For the next sentence, I struggled with the use of "equalized". Based on my understanding, would something more like "converted to _____ equivalents" be more appropriate?

    *AC: Yes, we used the observed generated TA and assumed this to be produced by denitrification. Then, we subtracted it of the produced $N_2$, and assumed the remaining $N_2$ to be produced by anammox. 1) For a better understanding, we can change the verb "combined" into "compared". 2) We can also change the next sentence to clarify its meaning in accordance to your suggestion.*

5. Section 3.2.2 discusses how the measurements at a specific location vary during an ebb tide. Currently, I find the information about this process study lacking, and it difficult for me to contextualize the results within the larger study. A few potential points to improve are specified below
    1. Line 230 specifies that the chemistry of the water column changed due to the tide. Can you provide a subfigure that includes the tidal height during sampling?
    2. I am concerned that some of the data provided here reflects conditions at other places in the watershed, which were advected into the sampling location. Would it be possible to plot the bathymetry near the sampling site? Also, on line 114, your calculation includes the distance and the change in time for sampling, but not the advection velocity. Can you report the water velocity during the sampling timeframe? I think that the RV *Ludwig Prandtl* should have an ADCP on it. Even an approximate value could help constrain transport in this region.

    *AC: Thank you for these suggestions. 1) Yes, we can provide a subfigure that includes the tidal height during sampling. 2) We don't have bathymetry data of the Ems and also no ADCP data. Even if we think that advection velocity is important, we are unable to quantify it here. However, advection is more present in the upper water column near the surface than in the deeper water column close to the bottom, i.e., in the fluid mud, where SPM is very high and*

*where we did the incubation experiments. As support, we can show the difference of the flow speed at the surface (0.94 m sec$^{-1}$) and the much slower flow speed near the bottom (0.05 m sec$^{-1}$).*

Other errata:

1. Would you mind adding a data analysis/software subsection to your methods? This is particularly relevant for figs 2c-d and 3b, because when there's uncertainty in both the x- and y-variables, a type 2 linear regression is required.

   *AC: We can add a small subsection with the data analysis / software we used. In particular, Fig. 2c-d and 3b were done with the original data, the RStudio software (1.3.1073), and the package "ggplot2".*

2. In figure 4, the default colormap is not perceptually uniform or accessible for many colorblind people. ODV has better options, such as any of the colormaps with just a single color or viridis. Can you change the colormaps here? See Crameri et al. (2020), "The misuse of colour in science communication" for more information.

   *AC: Thank you for this advice. We will try to adjust the ODV plot to a colorblindness friendly plotting.*

3. In figure 5, would you mind changing the markers for each series of your data? It's a bit hard to see with just the color changes. A different line type could also help.

   *AC: Yes, we can adjust this figure for more clarity.*

4. With regards to the carbon data in PANGAEA, please provide the DOI before the manuscript is finalized! I appreciate the authors' attempt at making this information accessible.

   *AC: The data are in preparation to be released.*

5. Line 50: I regret to inform you that this paper is probably one of the first, rather than the first, manuscripts to differentiate between canonical denitrification and anammox using carbon parameters. I believe that "Coincident Biogenic Nitrite and pH Maxima Arise in the Upper Anoxic Layer in the Eastern Tropical North Pacific" by Cinay and Dumit et al. (2022) (https://doi.org/10.1029/2022GB007470) uses these measurements for a similar purpose, even if their mathematics differ significantly. I would recommend slightly rephrasing this claim.

   *AC: Even if the paper you supposed is different to our approach, we can rephrase the claim into "one of the first".*

---

## Author Comment (AC2)

**RC2**: 'Comment on bg-2022-226', Anonymous Referee #2, 07 Jan 2023 reply

This is a review report for the manuscript, entitled: "Alkalinity and nitrate dynamics reveal dominance of anammox in a hyper-turbid estuary" by Norbisrath et al (Manuscript number: bg-2022-226). Overall, stable isotope analysis reveals that N2 production in tidal rivers and fluid slurries are contributed by denitrification and anammox processes, respectively, and discusses two nitrate/nitrite respiration pathways and their effects on TA production. However, the authors' focus seems to be on these two nitrate/nitrite respiration pathways, with only a small amount of text devoted to the contribution of nitrate/nitrite respiration pathways to TA. The experimental design idea is good, but it seems to be inconsistent with the focus of the article, and the authors are recommended to reorganize the story line. I do not think the manuscript can be published in its current form.

*Dear referee #2,*

*We thank you for your feedback. We do appreciate time and efforts, referees spent to assess our manuscript, independent of the overall outcome of such a comment.*
*However, admittedly, we feel that we are unable to respond to your the comments, in particular due to the brevity and untargeted nature of the comment. Also we are confident that by addressing referee #1 comments this story line will become more clear and we also will carefully consider reorganizing it.*

---

## Author Response (AR1)

**RC1**: 'Comment on bg-2022-226', Anonymous Referee #1, 01 Jan 2023 reply

This paper focuses on nitrogen cycling, specifically nitrogen loss mechanisms, within an estuary. By supplementing nitrogen measurements with carbon system measurements, the researchers estimated the amount of fixed nitrogen loss that occurred due to canonical denitrification versus anammox. High resolution sampling from a river endmember to the North Sea, incubations, and a process station provide detailed characterization of the study site. Unfortunately, I struggled with following several analyses in this paper, and I would appreciate the authors clarifying these sections, which in many cases involves presenting the data in a different fashion. I suggest major revisions, as specified below.

*AC: Dear referee #1, we thank you for your feedback and the helpful comments and suggestions. We addressed your suggestions in order to improve the manuscript.*

1. The primary conclusions of this paper hinge on comparing total alkalinity measurements and $N_2$ production rates in fluid mud samples. I did not entirely understand the conditions for these incubations as well as the specific mathematics for concluding that anammox is the dominant metabolism. Here are some specific challenges

   *AC: We have rearranged the material and methods part a bit to make the conditions and the idea behind the incubation clearer.*

   1. Line ~145: In the incubation set-up, the authors specify that fluid mud samples were taken from stations that were low in oxygen. Can you quantify its concentration? If the concentration is below the sensor detection limit, can you specify that and the limit you are using?

      *AC: Yes, we added the surface oxygen concentrations at this station, as well as the oxygen concentrations at the start of the incubations.*

   2. On this same idea, anaerobic respiration also increases total alkalinity. It is not clear to me how this paper differentiates the amount of alkalinity produced by aerobic respiration versus canonical denitrification. Perhaps the oxygen concentration is so low that the small amount of alkalinity produced it produces is neglible? Aerobic respiration does produce a small amount of alkalinity (see "An assessment of ocean margin anaerobic processes on oceanic alkalinity budget" by Hu and Cai (2011) for the precise mathematics).

      *AC: Because of the very low oxygen concentrations ($<60$ μM ($<2$ mg $L^{-1}$) at the start of the incubation and the decreasing oxygen concentrations during the incubation, we rather exclude aerobic respiration as source of alkalinity in the incubation experiment and the fluid mud.*

   3. The structure of Fig. 6 took me a long time to understand the information it contained. Perhaps a plot of DIC vs TA with two lines on it for each sample, and the caption saying that 43 hours elapsed? Another option could be two subplots of TA and DIC with time on the x-axis, as this could also be easier to parse. In addition, I think that it would be helpful to see the $N_2$ data over time for these incubations. Since these values are also averaged, can you add error bars? To visually represent the calculations performed and the link between

alkalinity and $N_2$ production, perhaps another figure plotting TA vs $N_2$ and then adding in vectors for the denitrification-derived $N_2$ and annamox-derived $N_2$ could improve clarity?

*AC: Thank you for these suggestions. We changed the style of the plot by adding a separation of the two stations, and by plotting time on the x-axis for a better visualization. A plot of DIC vs. TA is not useful since it would show another result. Because the TA/DIC and $N_2$ incubation were not in the same bottle, and we only have start and end values for the TA/DIC incubation, we think a combination of both as not useful, in particular, because the $N_2$ production of all $N_2$ species of these incubations is given in Fig. 5. Because we plotted each incubation $N_2$ pattern as a single line (Fig. 5), error bars are not necessary. However, the standard deviation of the averaged $N_2$ value that we used for this estimate is given in table 2.*

4. From the description of your methods (line 170), I anticipate that you calculated the amount of $N_2$ that would be produced based on the measured TA, then compared that TA-calculated $N_2$ against the measured $N_2$ to determine the annamox $N_2$. I would like some clarity that this is the method that was used, because some of the verb choice confused me.
    1. Saying "…we combined the TA generation… with the average $N_2$ production per station" suggests to me that these values were added. Would "compared" be a better verb than "combined"?
    2. For the next sentence, I struggled with the use of "equalized". Based on my understanding, would something more like "converted to _____ equivalents" be more appropriate?

*AC: Yes, we used the observed generated TA and assumed this to be produced by denitrification, subtracted it of the produced $N_2$, and assumed the remaining $N_2$ to be produced by anammox. We rearranged the methods section a bit to make it clearer. 1) For a better understanding, we changed the verb "combined" into "compared". 2) We also changed the next sentence to clarify its meaning in accordance to your suggestion.*

Section 3.2.2 discusses how the measurements at a specific location vary during an ebb tide. Currently, I find the information about this process study lacking, and it difficult for me to contextualize the results within the larger study. A few potential points to improve are specified below

    3. Line 230 specifies that the chemistry of the water column changed due to the tide. Can you provide a subfigure that includes the tidal height during sampling?
    4. I am concerned that some of the data provided here reflects conditions at other places in the watershed, which were advected into the sampling location. Would it be possible to plot the bathymetry near the sampling site? Also, on line 114, your calculation includes the distance and the change in time for sampling, but not the advection velocity. Can you report the water velocity during the sampling timeframe? I think that the RV *Ludwig Prandtl* should have an ADCP on it. Even an approximate value could help constrain transport in this region.

*AC: Thank you for these suggestions. 1) Yes, we provided a subfigure that includes the tidal height during sampling (Fig. 4j). 2) We don't have bathymetry data of the Ems and also no ADCP data. Even if we think that advection velocity is important, we are unable to quantify it here. However, advection is more present in the upper water column near the surface than in the deeper water column close to the bottom, i.e., in the fluid mud, where SPM is very high and where we did the incubation experiments. As support, we added the difference of the flow speed at the surface (0.94 m sec⁻¹) and the much slower flow speed near the bottom (0.05 m sec⁻¹).*

Other errata:

1. Would you mind adding a data analysis/software subsection to your methods? This is particularly relevant for figs 2c-d and 3b, because when there's uncertainty in both the x- and y-variables, a type 2 linear regression is required.

   *AC: Thank you for this advice. We corrected the analyses accordingly, and added a subsection (2.7 Data analysis) with the used data analyses / software.*

2. In figure 4, the default colormap is not perceptually uniform or accessible for many colorblind people. ODV has better options, such as any of the colormaps with just a single color or viridis. Can you change the colormaps here? See Crameri et al. (2020), "The misuse of colour in science communication" for more information.

   *AC: Thank you for highlighting it. Of course, we adjusted the ODV plot to a colorblind friendly color scheme.*

3. In figure 5, would you mind changing the markers for each series of your data? It's a bit hard to see with just the color changes. A different line type could also help.

   *AC: Yes, we adjusted this figure for more clarity. We used more intense colors and different line types.*

4. With regards to the carbon data in PANGAEA, please provide the DOI before the manuscript is finalized! I appreciate the authors' attempt at making this information accessible.

   *AC: The data are in preparation to be released. In case this process works faster, we will adjust the availability information. Until then, the data are available from the corresponding author upon request. This is also stated in the availability information.*

5. Line 50: I regret to inform you that this paper is probably one of the first, rather than the first, manuscripts to differentiate between canonical denitrification and anammox using carbon parameters. I believe that "Coincident Biogenic Nitrite and pH Maxima Arise in the Upper Anoxic Layer in the Eastern Tropical North Pacific" by Cinay and Dumit et al. (2022) (https://doi.org/10.1029/2022GB007470) uses these measurements for a similar purpose, even if their mathematics differ significantly. I would recommend slightly rephrasing this claim.

   *AC: We rephrased the claim into "one of the first".*

**RC2**: 'Comment on bg-2022-226', Anonymous Referee #2, 07 Jan 2023 reply

This is a review report for the manuscript, entitled: "Alkalinity and nitrate dynamics reveal dominance of anammox in a hyper-turbid estuary" by Norbisrath et al (Manuscript number: bg-2022-226). Overall, stable isotope analysis reveals that N2 production in tidal rivers and fluid slurries are contributed by denitrification and anammox processes, respectively, and discusses two nitrate/nitrite respiration pathways and their effects on TA production. However, the authors' focus seems to be on these two nitrate/nitrite respiration pathways, with only a small amount of text devoted to the contribution of nitrate/nitrite respiration pathways to TA. The experimental design idea is good, but it seems to be inconsistent with the focus of the article, and the authors are recommended to reorganize the story line. I do not think the manuscript can be published in its current form.

*AC: Dear referee #2, we regret to hear that the story line apparently was so unclear, and that there was such a need for refocusing the manuscript. To meet this criticism, we reformulated the abstract, introduction, material and methods, and conclusions sections slightly, hopefully underscoring the intention of our study. In fact, we wanted to show that the combination of TA and $N_2$ production measurements can be a powerful tool to disentangle nitrate respiration pathways. In the case of our study, it surprisingly showed that anammox dominated $N_2$ production, which was entirely unexpected in such a heterotrophic estuary.*

---

## Author Response (AR2)

BG2022-226 Reports for acceptance:

Report #1:
The authors did an excellent job addressing the concerns of previous reviewers, and this manuscript is far clearer. I noticed some typos in the first pages, and I encourage the authors to take another read through to fix those. Nevertheless, I appreciate the effort that went into this revision, and after a quick look through, this paper should definitely be published!

*AC: Thank you very much for this positive feedback. In order to correct the typos, we had the paper read by a native speaker.*

Report #2:
Based on the information provided, the article focuses on investigating the nitrate/nitrite respiration pathways and their impact on total alkalinity (TA) generation in the highly turbid estuary of the Ems River. The researchers sampled various locations and conducted incubation experiments to determine the dominant pathways for nitrogen gas ($N_2$) production. The findings suggest that in the water column of the tidal river, denitrification is the dominant pathway for $N_2$ production, as indicated by stable nitrate isotopes. However, in the fluid mud of the tidal river, the majority of $N_2$ production is attributed to anammox (anaerobic ammonium oxidation) rather than denitrification. This implies that anammox plays a significant role in the $CO_2$ storage capacity of the coastal waters adjacent to the estuary. The article highlights the importance of understanding the carbon cycle and nitrogen dynamics in coastal ecosystems, particularly in estuaries subject to human interventions and high nutrient inputs. The research contributes to our knowledge of the factors influencing $CO_2$ storage capacity and the relative contributions of different nitrogen respiration pathways in estuarine environments. Overall, the article provides valuable insights into the dominance of anammox in the studied estuary and its implications for carbon and nitrogen cycling. The findings contribute to our understanding of estuarine biogeochemistry and have implications for coastal management and climate change mitigation strategies. I think that it is an interest and worthwhile topic to be published in BG, but there are some problems to revise. A minor revision is required to make the paper more understandable.
*AC: Thank you very much for this positive feedback. We have improved the paper according to your suggestions.*

The comments are summarized below:
1) Line 52-53, the sentence "Based on TA generation and an isotope pairing approach, we find that anammox dominates in this heterotrophic environment" is suggested to be deleted, because this conclusion is from this study, which is not suitable to show in Introduction.
*AC: We deleted this sentence.*
2) Line 85, water samples dried at 550°C and weighed, how to do?
*AC: We corrected the temperature and added some clarification and the reference for the detailed sampling protocol.*
3) Line 99, "potentiometric titration" not "potentiometric"
*AC: Corrected.*